# Synchronization of complex human networks

Shir Shahal[1], Ateret Wurzberg[1], Inbar Sibony[1], Hamootal Duadi[1], Elad Shniderman[2], Daniel Weymouth[2], Nir Davidson [3] & Moti Fridman [1✉]

The synchronization of human networks is essential for our civilization and understanding its dynamics is important to many aspects of our lives. Human ensembles were investigated, but in noisy environments and with limited control over the network parameters which govern the network dynamics. Specifically, research has focused predominantly on all-to-all coupling, whereas current social networks and human interactions are often based on complex coupling configurations. Here, we study the synchronization between violin players in complex networks with full and accurate control over the network connectivity, coupling strength, and delay. We show that the players can tune their playing period and delete connections by ignoring frustrating signals, to find a stable solution. These additional degrees of freedom enable new strategies and yield better solutions than are possible within current models such as the Kuramoto model. Our results may influence numerous fields, including traffic management, epidemic control, and stock market dynamics.

[1] Faculty of Engineering and the Institute of Nanotechnology and Advanced Materials, Bar-Ilan University, 5290002 Ramat Gan, Israel. [2] Department of Music, Stony Brook University, Stony Brook, NY 11794, USA. [3] Department of Physics of Complex Systems, Weizmann Institute of Science, Rehovot, Israel. ✉email: mordechai.fridman@biu.ac.il

The synchronization of coupled ensembles appears in numerous fields, including biology[1–3], astronomy[4], psychology[5,6], optics[7–9], economics[10], and politics; at different size scales, from the synchronization of planets[4] to the synchronization of subatomic particles[11]; and in different time-scales, from slow-moving mechanical structures[12,13] to coupled ultrafast lasers[14,15]. Synchronization is crucial for the life of all living species on our planet[1,2], from the cellular level[16–18] to the crowd synchrony of large groups[19]. In particular, the synchronization of human networks is essential for our civilization[20–22] and can impact the physical and mental well-being of individuals in groups[5,6]. Understanding the motivations, behavior, and basic parameters that govern the dynamics of human networks is important for many aspects of our lives, including stock market dynamics[10], traffic management[23], epidemic control[24], and investigating the decision-making processes in different types of groups[25–29]. Additionally, studying the dynamics of human networks will help predict the consequences of introducing artificial intelligence into our highly connected world, where each node in a computer network will have complex decision-making ability[30,31].

Human ensembles and crowd synchrony[19] have been investigated in recent years. Synchronized brokers in the stock market were found to earn more money[10], the synchronization of crowd attention was shown to be a basic survival mechanism[32,33], pedestrians walking on the London Millennium bridge synchronized their footsteps through the bridge vibrations to form macroscopic oscillations of the bridge above a critical number[12], the collective movement of concert audiences showed vortexes and gas-like states[34,35], the synchronized movements of dancers differ from those of nondancers[36,37], music players are following each other according to their musical instrument[38–40], and an audience clapping hands shows both synchronization and period doubling[41,42]. Synchronization in the broader sense of coordinating decision-making between humans on complex networks has also been studied[43,44].

However, all these seminal studies had limited control over the network parameters, namely, the connectivity of the network, coupling strength, and delay between individuals, and were subject to noisy environments. In particular, these studies focused mostly on all-to-all coupling, whereas current social networks and human interactions are often based on complex coupling configurations. To date, there are no studies of synchronization of rhythmic behavior of humans in complex networks, for example, one-dimensional, two-dimensional, scale-free, or small-world connectivity in a controlled environment[45–47]. Additionally, the influence of changing the coupling strength or the delay between two individuals is critical for the dynamics of the network[48–50] and has not been studied in human networks thus far.

We study the synchronization between professional violin players in complex human networks with full and accurate control over the network connectivity, coupling strength of each connection, and delay between players. We set 16 isolated electric violin players to repeatedly play a musical phrase. We collect the output from each violin and control the input to each player via noise cancellation headphones. The players cannot see or hear each other apart from what is heard in their headphones. All the players start playing the first phrase with the help of an external rhythmical beat, to verify that they all start with the same playing period and phase. The rhythmical beat is stopped after the first phrase, and the only instruction to the players is to try to synchronize their rhythm to what they hear in their headphones. A picture of the experimental setup is shown in Fig. 1, and the musical phrase is shown in the inset. We establish different network connectivities and introduce delayed coupling between the players while monitoring the phase, playing period, volume, and frequency of each player with a mixing system. Our system is

the first for investigating human networks with full and accurate control over the network parameters, including, the connectivity, the coupling strength, and the delay of each connection. In addition, this is the first system where the parameters of the network can be changed in a controlled manner in real time, enabling the study of dynamical human networks.

Our results reveal that the usual models for coupled networks such as the Kuramoto model[51–54] cannot always be applied to human networks. We found that the players can change their playing period[3,41,42,55] and can delete connections by completely ignoring frustrating signals[56] to find a stable solution to the coupled network. These additional degrees of freedom enable new strategies and yield better solutions than are possible within the simple Kuramoto model. To analyze the dynamics of a human network and the influence of different parameters on its global behavior, we extended the Kuramoto model to take into account these important abilities of the human mind, which have been neglected thus far.

## Results

**Coupled violin players without delay**. In our first experiment, we set the coupling between the players to zero, causing the players to hear only themselves. We measure the time it takes for each player to play the musical phrase and denote this time as the playing period of the player, $T_i(t)$. In Fig. 2a, we show the phase of each player as a function of time, where blue denotes the beginning of the musical phrase and yellow denotes the end. In Fig. 2b, we show the playing period of all the players and the standard deviation of their period as a function of time. The opening phrase, accompanied by an external rhythmical beat, verified that all the players start with the same playing period; after the first phrase, the beat stopped, and the playing period of each player deviates towards the player's natural one. The playing periods of the players are spreading as a function of time, reflecting that the players cannot hear or see each other.

Then, we introduce coupling between the different players with our mixing system. The coupling strength is defined as the ratio between the volume of the coupled violin compared to the volume of the player's own violin while maintaining the total volume that each player hears constant. The volume level is monitored to make sure it stays within the linear response range of the human hearing[57]. We compare two configurations for the players, a one-dimensional open chain, which is a network with the lowest possible connectivity, and an all-to-all coupling, which is a network with the highest possible connectivity. In each configuration, we start with a coupling strength of 0.5 and reduce it linearly to zero over a period of 4 min. We measure the in-phase order parameter in the network as a function of the coupling strength and present the results in Fig. 2c. The in-phase order parameter is calculated by $<\cos(\varphi_i - \varphi_j)>$, where $\varphi_i$ is the phase of the $i$th player, $\varphi_j$ is the phase of its coupled neighbor, and we average over all connections. Similar to other networks, the order parameter of the all-to-all configurations remains high for lower coupling strength compare to the one-dimensional configuration. (The order parameter does not reach zero since the playing time is limited to 4 min to keep the players focused.)

**Two coupled violin players with delay**. Next, we set the coupling strength to 0.5, which is strong enough to ensure synchronization, as shown by Fig. 2c. Then, we impose a delay on the coupling between the players, starting from zero delay and increasing it linearly, according to $d(t) = 0.0332t$, where $d$ is the delay and $t$ is time, so after 120 s the delay equals to 4 s, which is the starting playing period of the musical phrase. The delay prevents the players from synchronizing with each other, which leads them to shift from an in-phase synchronization to other states of

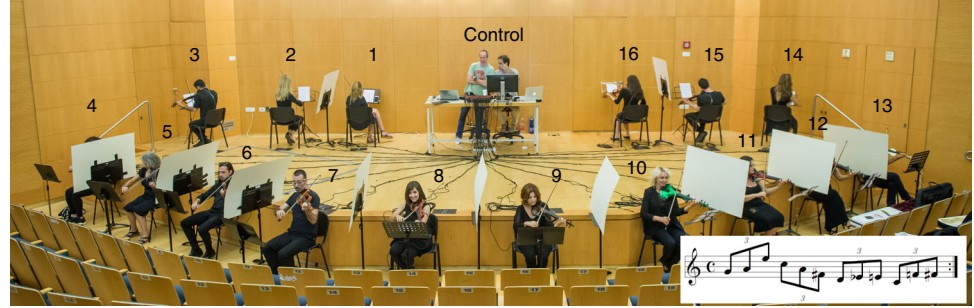

**Fig. 1 Sixteen coupled electric violin players repeating a musical phrase.** Sixteen violin players are playing with electric violins. The audio output from each violin is connected to our computer-controlled mixing system. Then, the mixing system sends to the headphones of each player a sum of audio signals of the desired connectivity, strength, and delay. All participants in the picture have approved publishing it. The musical phrase is presented in the inset.

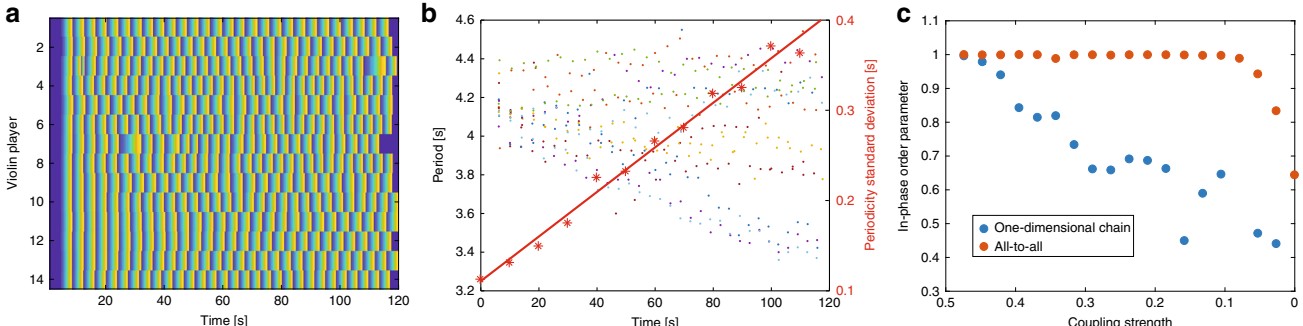

**Fig. 2 Uncoupled and coupled violin players in different configurations.** Here we show the playing period and the phase of each player as a function of time and the in-phase order parameter as a function of the coupling strength for different configurations. The playing period of each player denotes the time that it takes for each player to play the entire musical phrase, and the phase denotes where in the musical phrase the player is at a specific time. **a** The phase of each violin player as a function of time without coupling, where blue denotes the beginning of the musical phrase and yellow denotes the end. **b** The playing period of each player (color dots) and the standard deviation of the period (asterisks) with a fitted linear curve as a function of time without coupling, showing that with no coupling, each player is changing its playing period without any correlation to other players. **c** The in-phase order parameter of the network as a function of the coupling strength for two different configurations: one-dimensional chain with nearest-neighbor coupling and a global all-to-all coupling.

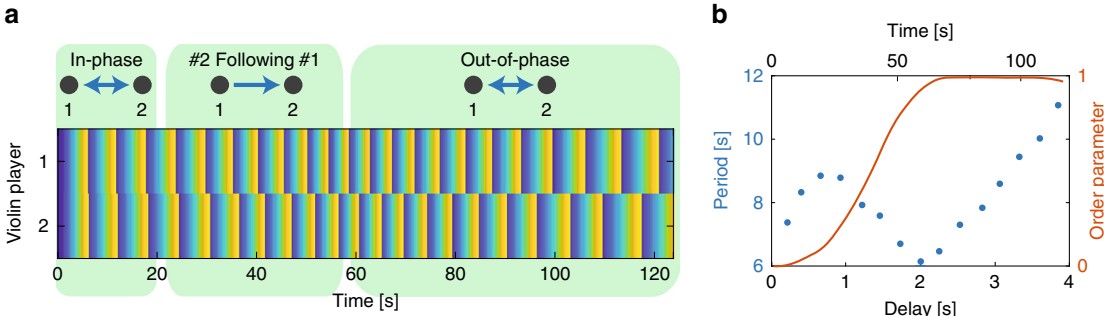

**Fig. 3 Two coupled violin players with a delay between them. a** The phase of each player along the musical phrase as a function of time in one representative measurement. When the delay is zero, the two players are following each other in-phase indicated by a double-head arrow between them. When the delay increases, only one of the players can follow the other, which is indicated by a single-head arrow. When the delay reaches half of the playing period time, the two players can follow each other again in an out-of-phase state of synchronization, indicated by a double-head arrow. **b** The mean playing period of all the players and the out-of-phase order parameter of the network as a function of the delay and time, averaged over a moving window.

synchronization[58]. We demonstrate these states of synchronization by examining the synchronization of two coupled violin players as a function of the delay, schematically shown in Fig. 3. In Fig. 3a, we present the phase of each player in the musical phrase by a color code as a function of time in one representative measurement. We determine that player $i$ is following player $j$ once they have the same playing period, namely $T_i(t) = T_j(t)$, and

their relative phase during at least one musical phrase, follows:

$$\varphi_j - \varphi_i = 2\pi \frac{d(t)}{T_i(t)}. \tag{1}$$

When Eq. (1) is satisfied, player $i$ is playing in synchrony with player $j$ as it sounds in its earphones. If Eq. (1) is not satisfied, even if the relative phase between them is constant in time, they

are not following each other. This can occur when both players are following a third player while ignoring each other.

In Fig. 3b, we show the averaged period of all the players as a function of the delay and time together with the out-of-phase order parameter, $< \sin(\varphi_i - \varphi_j)>$. The results reveal three states of synchronization: initially, the delay is zero, so the two players are perfectly synchronized in phase. With the introduction of the delay, they increase their playing period (play slower) to keep the delay small relative to the duration of each note. This state is emphasized on the left side of Fig. 3a and is indicated by the increased playing period, presented in Fig. 3b. This effect was also observed when playing over the Internet with a small delay[59]. When the delay is further increased, the players cannot maintain an in-phase synchronization state, as one of them starts to ignore the other and returns to its original playing period. In our case, player #1 ignores player #2, while player #2 still follows player #1, which is emphasized in the middle part of Fig. 3a. When the delay is increased to approximately half of the period, an out-of-phase synchronization emerges that satisfies both players, since, $\varphi_j - \varphi_i = \varphi_i - \varphi_j = 2\pi d(t)/T_{i,j}(t)$, so they are following each other. In this out-of-phase synchronization[58], when player $i$ is at the middle of the musical phrase, player $j$ is at the beginning or the end of the phrase, and vice versa. This state is highly stable; therefore, when the delay is further increased, the players increase their playing period to ensure that the delay is always half the playing period. This is shown in Fig. 3b, where the out-of-phase order parameter is presented by the red curve. Once this order parameter approaches unity, it stays there, and the playing period increases linearly with the delay. This is also observed by the checkerboard pattern on the right side of Fig. 3a.

To verify that the delay is changing slow enough, we measure the coupled violin players when the delay is changing at half the rate according to $d(t) = 0.0166t$ obtaining similar results. This indicates that, although the delay is constantly changing, the delay change-rate is slow enough so that at each point in time the network can be considered as quasistatic. In such a system, the players are not aware to the fact that the delay is changing and only react to its current value.

**Even number of coupled violin players**. When increasing the number of the coupled violin players to 4, 6, or 8, as shown in

Fig. 4a, d, and g, they follow the same behavior as the delay is increased: we first observe an in-phase synchronization with an increase in the playing period; next, each player spontaneously decides to ignore one of its inputs. In this stage, we observe two states of synchronization, a vortex state or an arrowhead state. If all the players ignore the same side and follow the other side, they create a vortex state of synchronization where the phase increases monotonically, as seen in Fig. 4h, while if some players are choosing to follow the player on one side and other players are choosing to follow the player on the another side, they create an arrowhead-shaped state of synchronization, as seen in Fig. 4e. Finally, when the delay reaches approximately half of the average playing period, a stable and highly ordered state of out-of-phase synchronization emerges, as evident by the checkerboard pattern emphasized at the right side of Fig. 4b, e, and h, together with the linear increase in the average playing period as a function of the delay and the out-of-phase order parameter, which approaches unity, as seen in Fig. 4c, f, and i. These results are identical whether the players are organized in open- or close-chain configurations. In the case of eight players, we also observe that the players are divided into two clusters, players 1–3 and players 4–7, while player 8 is somewhere between them[60]. The second cluster finds the out-of-phase synchronization state faster compared to the first cluster, so the dynamic of the second cluster is shown in Fig. 4i.

**Odd number of coupled violin players**. The total accumulate phase for an even number of violin players in a state of out-of-phase synchronization is an even integer multiplied by $\pi$, and is therefore consistent with the periodic boundary conditions of the loop. For odd numbers of violin players, this is not the case, and therefore the state of out-of-phase synchronization is no longer a stable solution[61–64]. In such cases, the players spontaneously choose to ignore one of the connections, which break the chain and forms an open chain where the out-of-phase synchronization state is possible. Thus, the players change the connectivity of the configuration into one with a stable solution. In Fig. 5, we present the results for three and five coupled violin players. When the delay is low, the players remain in an in-phase synchronization, as shown on the left side of 5a, c, while increasing the playing period, as shown in 5b, d. When we increase the delay, the players

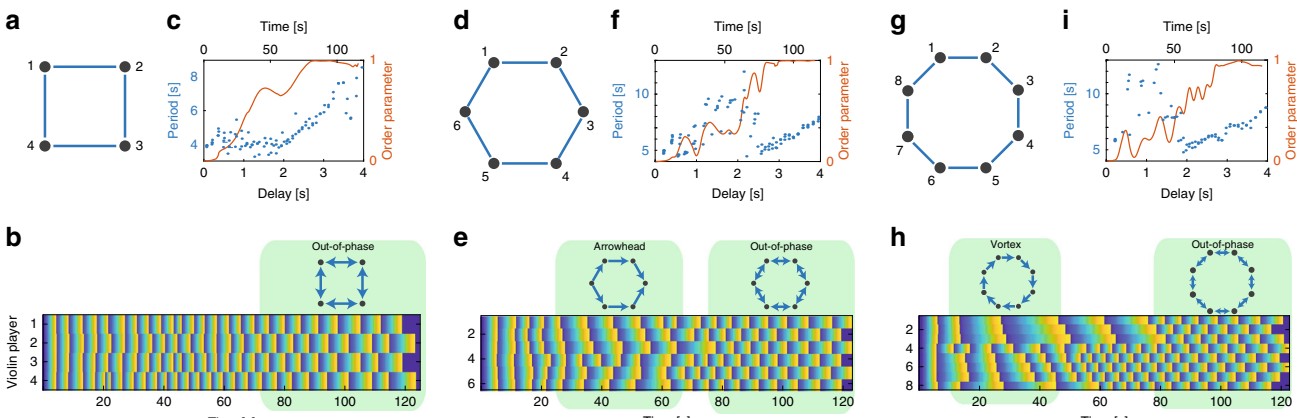

**Fig. 4 Four, six, and eight coupled violin players.** The configurations of the coupled players are schematically shown in **a**, **d**, and **g**. The phase of each violin player along the musical phrase as a function of time, in one representative measurement, is shown in **b**, **e**, and **h**. The delay between the players is increasing linearly in time. When the delay is low, we observe an in-phase synchronization; when the delay increases, we observe a vortex or an arrowhead state of synchronization; and when the delay is half of the playing period, we observe a stable out-of-phase synchronization. In **c**, **f**, and **i**, we present the playing period and the out-of-phase order parameter as a function of time and the delay. As shown, when the players experience out-of-phase synchronization, indicated by an order parameter of unity, the playing period increases linearly with the delay, remaining twice the delay to preserve the highly stable state of out-of-phase synchronization.

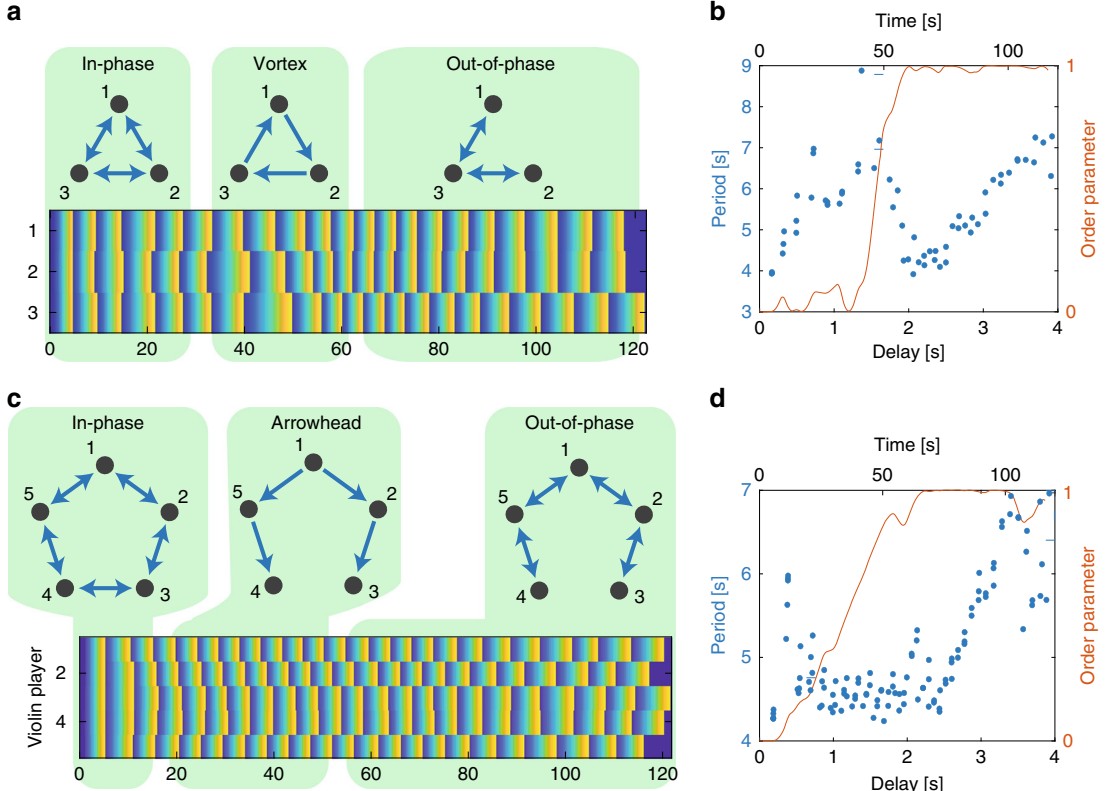

**Fig. 5 Three and five coupled violin players.** In the configurations of three and five coupled players, the out-of-phase synchronization state is no longer stable. In these cases, the players change the connectivity and ignore one of the links, reducing the system to an open chain where out-of-phase synchronization is stable again. **a** Three coupled players showing in-phase, vortex, and out-of-phase states of synchronization. **b** The playing period and the open-chain out-of-phase order parameter as a function of time and the delay, showing that once the players obtain an out-of-phase synchronization state, they maintain it by increasing the playing period with the delay. **c** Five coupled players, showing in-phase, arrowhead, and out-of-phase states of synchronization. **d** The playing period and the open-chain out-of-phase order parameter, showing the same behavior as the three coupled players.

choose either a vortex state, as shown in 5a, or an arrowhead state, as shown in 5c. When the delay reaches half of the playing period, the players prefer the state of out-of-phase synchronization while ignoring one of the connections, as shown on the right side of 5a, c. When this state is achieved, it is highly stable, as seen by the out-of-phase order parameter shown in 5b, d calculated for open-chain connectivity. When we increase the delay further, the players increase their playing period, keeping it twice the delay, to maintain the out-of-phase synchronization state, as shown in 5b, d, and similar to the dynamics of configurations with even number of players.

For nine or more coupled players, the violin players can find an approximate out-of-phase synchronization state without breaking the connection by shifting each player by $2\pi/9$ in addition to the out-of-phase synchronization. The combination of an out-of-phase with a vortex states is shown on the right side of Fig. 6a. We evaluate the out-of-phase order parameter, which reaches 0.9 instead of a unity due to this vortex shown in Fig. 6b. Nevertheless, this state is as stable as the regular out-of-phase states, as evident by the increasing playing period as a function of the delay while keeping the order parameter at 0.9. Here, similar to eight violin players, the players divided into two clusters, where one cluster found the out-of-phase synchronization state faster compared to the other[60]. In Fig. 6b, we show both clusters where the playing period of players 1–3 and 9 is denoted by the yellow dots and the playing period of players 4–8 is denoted by the blue dots. We see that when both clusters found the state of out-of-phase synchronization, they converged into a single cluster.

**Two-dimensional lattices configurations.** Finally, we measure the synchronization of the players when arranging them in a square and a triangular lattice configurations while increasing the delay. During the experiment, we monitor the relative phase between each pair of players and determine if they are coupled or not similar to the method described for the one-dimensional configurations and according to Eq. (1). The results are shown in Fig. 7, where the measured results of the square lattice are shown in Fig. 7a and the measured results of the triangular lattice are shown in Fig. 7b. When the delay is low, the players of the square lattice configuration are synchronized in phase, and when we increase the delay, they create a vortex states until reaching the state of out-of-phase synchronization, which is a stable solution for the square lattice configuration. In the triangular configuration, the players start with in-phase synchronization, and when we increase the delay, they cannot find a stable solution[61,65], so they ignore some of the connections and reduce the connectivity of the network to one based on square motifs or open chains. A reduced network that is based on square motifs or open chains is following the same dynamics as any chain with even number of players, and thus, can find the highly stable state of out-of-phase synchronization. This result is shown by the reduced network on the right side of Fig. 7b. When repeating the experiment, the players converge to a different solution every time, as shown in Figs. 7c–e, presenting solutions that include rings of four and six players and the breaking of the network into smaller coupled clusters. Once the players find a stable solution they tend to stay in it, while in some rare cases they switch from one stable solution to another.

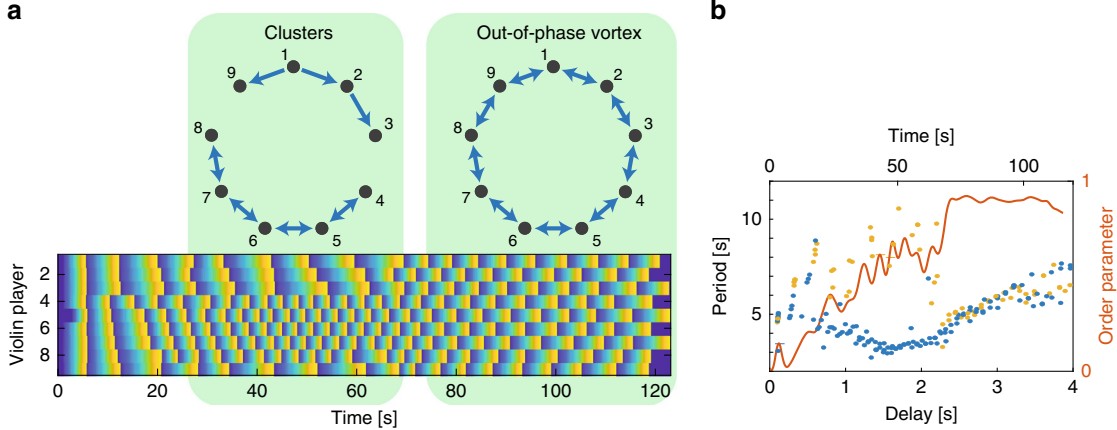

**Fig. 6 Nine coupled violin players illustrating an out-of-phase vortex state. a** The phase of each player along the musical phrase in one representative measurement as a function of time. Here we see the separation into two clusters, which combine into a single cluster with the checkerboard pattern indicating an out-of-phase synchronization state. **b** Playing period and out-of-phase order parameter as a function of time and the delay. The order parameter reaches 0.9 due to the vortex, but stays there while increasing the playing period as a function of the delay, indicating a stable state. Yellow dots —the playing period of players 1–3, and 9; blue dots—the playing period of players 4–8.

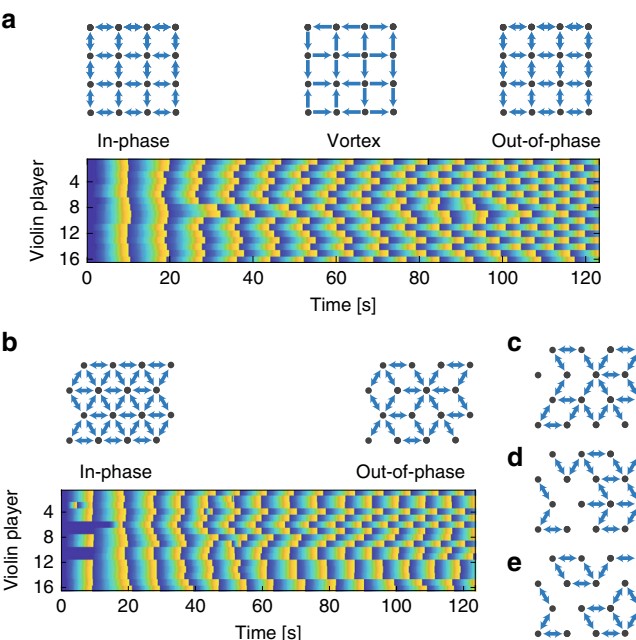

**Fig. 7 Sixteen violin players arranged in square and triangular configurations. a** The evolution of the square lattice configuration as a function of time showing in-phase, vortex, and out-of-phase states of synchronization. **b** The evolution of the triangular configuration showing that for the out-of-phase synchronization state, the connectivity of the network is reduced in order to reach a stable solution. We repeat the experiment and obtain a different solution each time, based on the same motifs. Three representative solutions are shown in **c–e**.

**Numerical models**. To develop a model for coupled human networks, we extend the simple Kuramoto model for coupled oscillators[51–54] to include broad-bandwidth oscillators and the ability of each oscillator to ignore some of the connections. We start by simulating coupled violin players with ring-like connectivity according to:

$$\frac{\partial \varphi_i}{\partial t} = \omega_i + \kappa \sum_j \sin\left(\varphi_j(t - \Delta t) - \varphi_i(t)\right), \qquad (2)$$

where $\varphi_i$ is the phase of the $i$th violin player, $\omega_i$ is the

eigenfrequency of the player, $\kappa = 0.2$ is the coupling strength, and $\Delta t$ is the delay between the players. We simulate the dynamics of different numbers of violin players and study the phase of each player compared to the others. We randomly choose the eigenfrequencies between $\omega = 0.25$ and 0.3 Hz with a uniform distribution, corresponding to a playing period of 3.3–4 s. We set the delay as a function of time according to $d(t) = 0.0332t$, so after 120 s the delay reaches 4 s. Representative results of four coupled players are shown in Fig. 8a. At first, the players are coupled in phase, and as the delay increases, the playing period likewise increases until a state of out-of-phase synchronization is achieved. This is also shown by the out-of-phase order parameter, which approaches unity at a delay of ~2 s. However, since the oscillators are narrow band, they cannot shift their playing period by more than 15%. Therefore, the players cannot maintain the out-of-phase state of synchronization when the delay is farther increased. Indeed, at a delay of ~3 s, the players leave this state and return to the state of in-phase synchronization. These results do not agree with the measured results, where the players adjust their playing period by up to a factor of 3 to maintain the state of out-of-phase synchronization.

We assume that humans have broad bandwidth, which enables them to change their playing period over a wide range[3,66]. To include this broad bandwidth of humans in the model, we added an imaginary parameter to Eq. (2) as follows:

$$\frac{\partial \varphi_i}{\partial t} = \omega_i + \kappa \sum_j \sin\left(\varphi_j(t - \Delta t) - \varphi_i(t)\right) + \eta, \qquad (3)$$

where $\eta$ is the bandwidth factor. This parameter serves as an imaginary frequency leading to exponential decay in time. Therefore, it lowers the Q-factor of the cavity and increases the bandwidth. We repeated the simulations with $\eta = 1i$ and present the results in Fig. 8b. These results are in a better agreement with the measured results of even number of players than the simple Kuramoto model. The players find the out-of-phase synchronization state and maintain it by changing their playing period linearly with the delay. This is also evident by the order parameters, which remains close to unity.

For odd numbers of players, the Kuramoto model failed to reproduce the measured results and showed only vortex states of synchronization[61,65]. Representative results for three coupled violin players are shown in Fig. 9, where Fig. 9a shows three

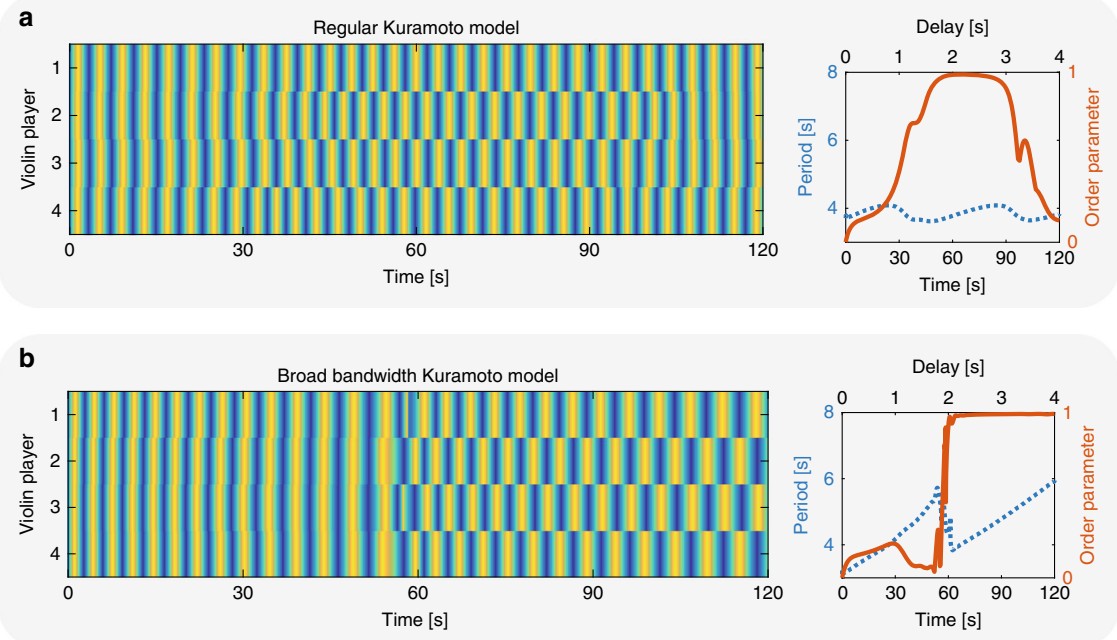

**Fig. 8 Calculated results of four coupled violin players. a** Calculated results of the regular Kuramoto model for four coupled players. In this model, the players cannot maintain the out-of-phase state of synchronization when the delay increases. **b** Calculated results of the broad-bandwidth Kuramoto model for four coupled players. In this model, the players change their playing period to maintain the out-of-phase state of synchronization.

coupled players with the regular Kuramoto model, and Fig. 9b shows with the broad-bandwidth Kuramoto model. Indeed, the players do not find the out-of-phase synchronization state, which is frustrated in three coupled players, as evident by the order parameter, which does not exceed 0.8.

Therefore, we extend the model to include the ability to delete contradicting connections. For any player with contradicting inputs we replace the sum in Eq. (3), with one neighbor. Representative results are shown in Fig. 9c. Here, we see that although the number of players is odd, the players find the out-of-phase synchronization state by ignoring one of the links. In this case, they ignore the connection between player 1 and player 3. This extended model agrees with the measured results for odd numbers of coupled violin players.

We compare three different strategies for choosing which connections to keep when a player encounter contradicting inputs from several coupled neighbors: keeping similar playing period, keeping similar phase, or choosing in random. In keeping similar playing period, the player follows the coupled players with closer playing period to its own. In keeping similar phase, the player follows the coupled players with closer phase to its own. In the random strategy, the player randomly chooses which player to keep and which to delete regardless of their phase or playing period. We simulate the dynamics of a triangular network of coupled players when we start with zero delay and linearly increase it. With all three strategies, the system finds an out-of-phase synchronization states by deleting connections and reducing the network connectivity to one based on motifs with an even number of players. We present typical reduced networks in Fig. 10 following each of the three different strategies. These calculated results reveal that all three strategies lead to the same dynamics. As long as each player can delete connections, the network changes its connectivity until finding a stable out-of-phase synchronization state. Therefore, the specific strategy each player has for choosing which inputs to follow, has no role in the macroscopic network dynamics of coupled violin players.

## Discussion

To conclude, we investigate the synchronization of rhythmic behavior of humans in networks with different types of connectivity where all the parameters of the networks are under control. We measure the phase and synchronization of coupled violin players in different network configurations and when introducing delay between the coupled players. We discover that human networks differ from previously studied networks in the ability of each player to adjust its playing period and to change the network connectivity by ignoring a coupled player and effectively deleting the connection. This ability serves as a unique and efficient mechanism to remove frustrating signals that hinder synchronization. When we couple an even number of players on a ring, the players find a stable out-of-phase synchronization state and tune their playing period accordingly as the delay increases. When we couple an odd number of players on a ring, the players change their connectivity and then adjust their playing period. We conclude that, the ability of human to identify conflicts in inputs and to adjust their response accordingly, which is well known[56], leads to unique dynamics when situated in networks. This research may impact numerous fields, including economics, decision-making research, epidemic spreading, information transfer modeling, traffic control, and more.

## Methods

**Experimental setup**. We set 16 isolated electric violin players to repeatedly play a musical phrase. The players play on Armando VL-D810. We collect the output from each violin into the Focusrite ClarettOctoPre sound system and control with a MAX/MSP software. The players cannot see or hear each other apart from what is heard in their noise cancellation headphones, Shure SE-215, which are connected to the output of the sound system. During the experiment, we record all 16 players with the 16-channel sound system.

**Composing the musical phrase**. The notes in the musical phrase were chosen while taking several considerations into account. First, it is important that different notes do not repeat for making it easier for the player to recognize where their coupled players are located in the musical phrase. Second, for easier analysis by preventing mixing with overtones, we keep the entire musical phrase at the same octave. Finally, we aim for a cyclic musical phrase without a clear beginning; therefore, a simple arpeggio is

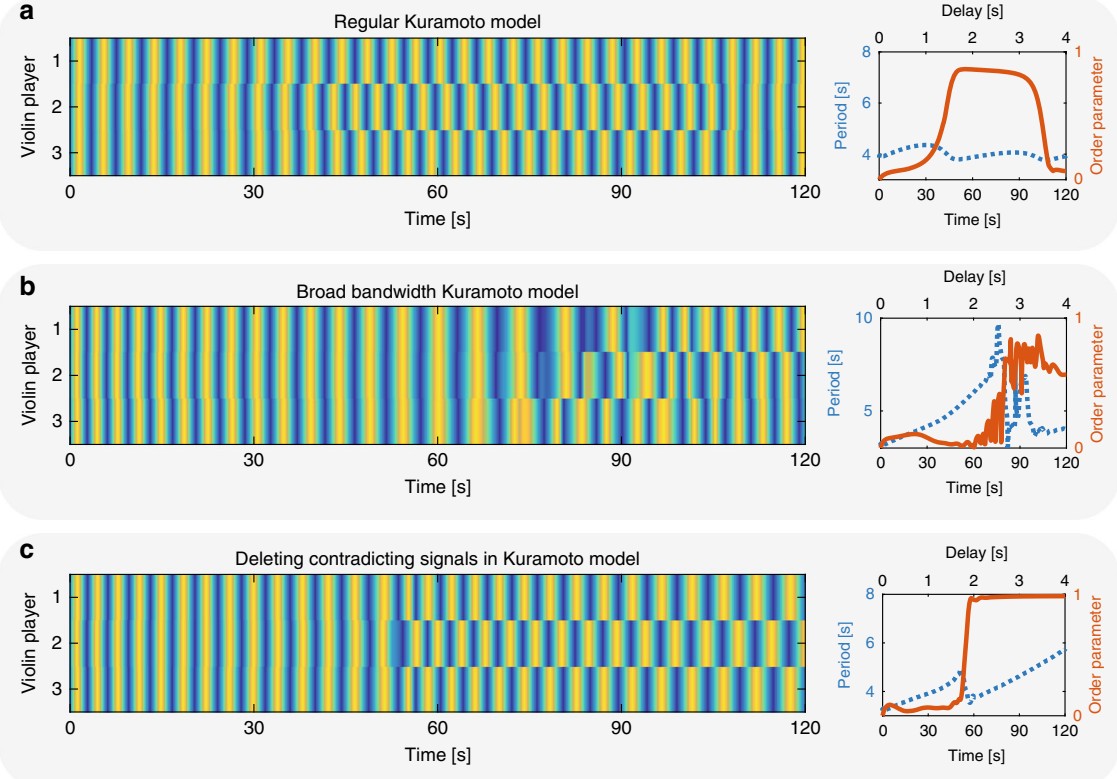

**Fig. 9 Calculated results of three coupled violin players. a** Regular Kuramoto model showing that the players can only reach a partial state of out-of-phase synchronization and cannot stay in that state when increasing the delay. **b** Broad-bandwidth Kuramoto model showing that the players can stay in the partial out-of-phase state of synchronization, but it is not stable due to frustration. **c** Broad-bandwidth Kuramoto model where each player can choose which connections to ignore. Here the players find the state of out-of-phase synchronization and remain in it while increasing the playing period. We calculate the order parameter in this situation according to the connections which the players are choosing to follow.

|  | Phase | Periodicity | Random |
|---|---|---|---|
| Simulation #1 |  |  |  |
| Simulation #2 |  |  |  |
| Simulation #3 |  |  |  |

**Fig. 10 Typical reduced triangular arrays.** Here, we show the connections that the system decides to keep from a triangular array in nine representative simulations. We compare three strategies for choosing which player to follow and which to ignore: phase—following connections with similar phase; periodicity—following connections with similar playing period; random—randomly choosing which connections to follow and which to ignore. In all cases, the system finds the out-of-phase synchronization state by reducing the network to a network based on motifs with an even number of players, indicating that the specific strategy for choosing which connections to keep has no role in the network dynamics.

not suitable. Nevertheless, we repeat all the experiments with other musical phrases and obtain similar results to verify our findings.

**Data analyzing**. The output file is analyzed off-line in Matlab by Fourier transforming the signal in a moving window of 100 ms, which allows us to identify the different notes and the timing of each note in addition to performing a manual

consistency check. Next, we calculate the playing period of each player and its location during the musical phrase, which is the player phase. By comparing the phase between two coupled players, we determine if they are following each other, if one is ignoring the other, or if both of them are ignoring each other. We determine that a connection between two players is maintained when the phase difference between them is equal to the delay over the playing period, according to Eq. (1).

We performed three full experimental sessions on three different dates and two more partial experimental sessions on two other dates. During each session, we repeat every configuration up to 4 times. Since we have 16 players, we repeat the same configuration with different players during the same run. Therefore, configurations of 2, 3, 4, 5, and 6 players are repeating 8, 5, 4, 2, and 2 times during the same run with different violin players, accordingly. In each experiment, we usually find faulty configurations that cannot be used due to either earphone malfunctioning, software problems, or players who did not understand the instructions and ignored what they heard in their earphones. It is easy to identify these faulty configurations when a player is playing without synchronization even when the coupling strength is high and there is no delay.

**Participants' consent**. All players signed a participant consent form to take part in the research and agreed to the use of all the data and pictures.

**Third-party images or previously published figures**. We confirm that our manuscript does not contain any third-party images or any previously published figures.

## Data availability
The datasets generated during the experiments, the analyzed data generated during the current study, and the data generated by the numerical simulations are available online at https://figshare.com/projects/Synchronization_of_complex_human_networks/81590.

## Code availability
The code for analyzing the data, the numerical simulation code, and the code for performing the experiments are available online at https://figshare.com/projects/Synchronization_of_complex_human_networks/81590.

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

## Acknowledgements
We thank the Joseph Fetter Museum of Nanotechnology in the Institute of Nano-technology and Advanced Materials at the Bar-Ilan University for supporting this research.

## Author contributions
S.S. analyzed the results and composed the musical phrase, A.W. coordinate the violin players, I.S. organized the location for the experiments, H.D. helped in coordinating all the participants and equipment, E.S. wrote the computer code for running the experiment, designed and installed the sound system, and performed the experiments, D.W. supervised over the musical part of the research, N.D. helped in designing the experiment, analyzing the results, and writing the manuscript, and M.F. conceived the idea, supervised over all the experiments, analyzed the results, and wrote the manuscript.

## Competing interests
The authors declare no competing interests.
