## [Peer Review File · Nature Communications]

REVIEWER COMMENTS

Reviewer #1 (Remarks to the Author):

This paper presents a careful and well-thought-out experiment and analysis of humans attempting to synchronize their individually periodic behavior --- in this case, violinists cyclically repeating an assigned musical phrase. A very striking general result is the extent to which this human activity is shown to yield distinct kinds of activity reminiscent of behaviors of physical and biological systems of much greater simplicity. Transitions between these distinct kinds of behavior ("dynamical phase transitions") are cleanly demonstrated and explained. Moreover, the paper presents an innovative experimental implementation and study of the effects of network structure and time delay along links. One of the more striking results is the discovery of a phenomenon whereby, under appropriate circumstances, a human effectively breaks a communication link from one of its neighbors in order to maintain sync with another neighbor. This work will be of great general interest as a paradigm for the study of synchronized behavior of groups of humans and of other complex systems. The authors may be interested in citing the following additional reference: E. Ott and T.M. Antonsen, Jr., "Frequency and Phase Synchronism in Large Groups: Low Dimensional Description of Synchronized Clapping, Firefly Flashing, and Cricket Chirping," CHAOS, vol.27, 051101 {2017}.

Reviewer #2 (Remarks to the Author):

I found the experiments and results to be moderately interesting, especially the adaptations that humans make to maintain synchronization under challenging conditions. I say only "moderately" because the task itself (repeatedly playing a single musical phrase) is so contrived and artificial; consequently, it's unclear how far the results will generalize, but still, this is a useful and original study of human rhythmic synchronization and its dependence on delay and coupling patterns. It's also notable that the authors found a generalization of the Kuramoto model that could account for the main results. The claim that the precise mechanism for ignoring frustrating interactions did not matter (any of three mechanisms will do) was surprising and interesting. I can well imagine this paper stimulating further work, both experimentally and theoretically, so I would recommend publication.

Two suggestions:

1) In the schematic diagrams of the network structures (Figures 3-7), I could not make sense of the arrows along the edges between players. What does a single directed arrow mean? Or a bidirectional double arrow? Does an arrow pointing from player a to player b mean that player a is phase-leading player b (and if so, by what amount of phase difference, or is that irrelevant to the arrow notation)? In any case, the arrow notation needs to be explained in the main text and in the figure captions.

2) In the introduction, the authors write, "To date, there are no studies of the synchronization of complex human networks". This seems a bit of an overstatement. The claim is perhaps defensible if synchronization is taken in a narrow sense to mean synchronization of *rhythmic* behavior. But synchronization in the broader sense of consensus and coordination has been studied experimentally by Michael Kearns and collaborators for human decision-making tasks on complex networks, e.g.,

Behavioral dynamics and influence in networked coloring and consensus

Stephen Judd, Michael Kearns, Yevgeniy Vorobeychik

Proceedings of the National Academy of Sciences Aug 2010, 107 (34) 14978-14982; DOI:

10.1073/pnas.1001280107

An Experimental Study of the Coloring Problem on Human Subject Networks, by Michael Kearns, Siddharth Suri, and Nick Montfort, Science 11 Aug 2006: Vol. 313, Issue 5788, pp. 824-827
DOI: 10.1126/science.1127207

In a related vein, there are potentially relevant studies on the "wisdom of the crowd" under different network conditions, in work by Damon Centola and others, e.g.,

Network dynamics of social influence in the wisdom of crowds
Joshua Becker, Devon Brackbill, and Damon Centola
PNAS June 27, 2017 114 (26) E5070-E5076

Reviewer #3 (Remarks to the Author):

This paper presents an interesting experimental study of synchronization in a human network of interacting violin players. I had reviewed an early version of this paper. The following points remain to be addressed before a publication recommendation can be made.

1. Why is the playing period of each player $T_i(t)$ a function of time? How is it measured exactly?
2. It would be better to introduce an equation to define the coupling strength. What does it mean "... and according to a logarithmic scale"?
3. Often the terminology used is not explained as for the cases of "crowd synchrony" or "out-of-phase synchronization". One cannot expect the broad readership of NC to be familiar with technical terms like these. The other major problem is that complex concepts such as "vortex synchronization" or "arrowhead synchronization" are introduced by looking at figures. I suggest that these states are first defined and then illustrated in figures.
4. The experiment about increasing the communication delay is interesting. However, I have the following questions: "the players are able to perceive an increase of the communication delay while they are playing. Correct? So they may be able to figure what the actual delay is as time progresses. If yes, then I would argue a control experiment is needed in which the delay is static and does not change in time.
5. One of the main findings is the emergence of an effective network coupling topology for which certain links are neglected/ignored. However this could be better explained. How is the effective topology determined? Is it time invariant? Are the connections of this topology always bidirectional? Also, it appears some of these preferred emerging topology are "square motifs or open chains" -- is there an explanation for this? An alternative counter-explanation for neglecting some of the links, which would find excellent analogies in the studies on coordinated motion of animals, is that the human/animal mind can only keep track of a limited number of "signals" at the same time.

6. " Therefore, the specific psychology of each player has no role in the macroscopic network dynamics of coupled violin players" not sure psychology is the right word here?

7. It's good that the paper presents a simple model that attempts to explain the experimental observations. However, I do not understand how in the model the three different mechanisms considered for deleting links can all lead to the same dynamics. Do the authors have a clear understanding of this? Also, how are the frequencies of the Kuramoto oscillators and all the parameters (eg coupling strengths, η , etc) chosen? An alternative model could include a mechanism for network adaptation in which the coupling strengths are tuned in order to maintain synchronization, see Physical Review E 79 (1), 016201 2009 and CHAOS 20 (1), 013103 2010.

8. What do the authors exactly mean by "adjust the periodicity"?

9. This sentence from the conclusions is a bit vague: " Our system will be extended to investigate decision making models in different configurations, bifurcation and phase transition in human networks, and the nonlinearity of crowds. "

REVIEWER COMMENTS

Reviewer #1 (Remarks to the Author):

“This paper presents a careful and well-thought-out experiment and analysis of humans attempting to synchronize their individually periodic behavior --- in this case, violinists cyclically repeating an assigned musical phrase. A very striking general result is the extent to which this human activity is shown to yield distinct kinds of activity reminiscent of behaviors of physical and biological systems of much greater simplicity. Transitions between these distinct kinds of behavior (“dynamical phase transitions”) are cleanly demonstrated and explained. Moreover, the paper presents an innovative experimental implementation and study of the effects of network structure and time delay along links. One of the more striking results is the discovery of a phenomenon whereby, under appropriate circumstances, a human effectively breaks a communication link from one of its neighbors in order to maintain sync with another neighbor. This work will be of great general interest as a paradigm for the study of synchronized behavior of groups of humans and of other complex systems.”

We thank the reviewer for the positive comment.

1. *“The authors may be interested in citing the following additional reference:*

E. Ott and T.M. Antonsen, Jr., "Frequency and Phase Synchronism in Large Groups: Low Dimensional Description of Synchronized Clapping, Firefly Flashing, and Cricket Chirping," CHAOS, vol.27, 051101 (2017).”

We thank the reviewer for noticing us about this interesting paper and we added it as Reference [3] to the introduction and to paragraph 6 when discussing the model.

Reviewer #2 (Remarks to the Author):

“I found the experiments and results to be moderately interesting, especially the adaptations that humans make to maintain synchronization under challenging conditions. I say only “moderately” because the task itself (repeatedly playing a single musical phrase) is so contrived and artificial; consequently, it’s unclear how far the results will generalize, but still, this is a useful and original study of human rhythmic synchronization and its dependence on delay and coupling patterns. It’s also notable that the authors found a generalization of the Kuramoto model that could account for the main results. The claim that the precise mechanism for ignoring frustrating interactions did not matter (any of three mechanisms will do) was surprising and interesting. I can well imagine this paper stimulating further work, both experimentally and theoretically, so I would recommend publication.”

We thank the reviewer for the positive comment and for the recommendation to publish our manuscript. The reviewer is correct that our system is artificial, however, we used professional violin players, and these players are trained to synchronize to what they hear while playing

repeatedly musical phrases. Therefore, our experiment can serve as a first step to understand how humans behave under situations they are well trained for (musical improvisation and its “synchronization” could be a very interesting extension of our current work and connect to more general human behavior). We revised the manuscript according to all the reviewer specific suggestions.

- 1) *“In the schematic diagrams of the network structures (Figures 3-7), I could not make sense of the arrows along the edges between players. What does a single directed arrow mean? Or a bidirectional double arrow? Does an arrow pointing from player a to player b mean that player a is phase-leading player b (and if so, by what amount of phase difference, or is that irrelevant to the arrow notation)? In any case, the arrow notation needs to be explained in the main text and in the figure captions.”*

The reviewer is correct. The meaning of a single directed arrow means that player a is phase-leading player b by the delay between them. Per the reviewer comment, we added this explanation to Fig. 3, and revised figures 4, and 6, so they will be consistent with this. The revised figure, and figure caption:

Fig. 3. Two coupled violin players with a delay between them; (a) The phase of each player along the musical phrase as a function of time in one representative measurement. When the delay is zero, the two players are following each other in-phase denoted by a double-head arrow between them. When the delay increases, only one of the player can follow the other which is indicated by a single-head arrow. When the delay reaches half of the period time, the two players can follow each other again in out-of-phase state of synchronization, indicated by a double-head arrow. (b) The mean period of all the players and the out-of-phase order parameter of the network as a function of the delay and time, averaged over a moving window.

We also revised all the arrows in figures 4 and 6 to follow the same logic.

- 2) *“In the introduction, the authors write, “To date, there are no studies of the synchronization of complex human networks”. This seems a bit of an overstatement. The claim is perhaps defensible if synchronization is taken in a narrow sense to mean synchronization of *rhythmic* behavior.”*

Per comment by reviewer, we revised the introduction in paragraph 3:

“To date, there are no studies of synchronization of rhythmic behavior of humans in complex networks, e.g., one-dimensional, two-dimensional, scale-free or small-world connectivity in a controlled environment.”

And in the conclusions:

“To conclude, we investigated the synchronization of rhythmic behavior of humans in networks with different types of connectivity where all the parameters of the networks are under control.”

- 3) *"But synchronization in the broader sense of consensus and coordination has been studied experimentally by Michael Kearns and collaborators for human decision-making tasks on complex networks, e.g.,*

Behavioral dynamics and influence in networked coloring and consensus Stephen Judd, Michael Kearns, Yevgeniy Vorobeychik Proceedings of the National Academy of Sciences Aug 2010, 107 (34) 14978-14982; DOI: 10.1073/pnas.1001280107

An Experimental Study of the Coloring Problem on Human Subject Networks, by Michael Kearns, Siddharth Suri, and Nick Montfort, Science 11 Aug 2006: Vol. 313, Issue 5788, pp. 824-827 DOI: 10.1126/science.1127207

In a related vein, there are potentially relevant studies on the “wisdom of the crowd” under different network conditions, in work by Damon Centola and others, e.g.,

Network dynamics of social influence in the wisdom of crowds Joshua Becker, Devon Brackbill, and Damon Centola PNAS June 27, 2017 114 (26) E5070-E5076"

We thank the reviewer for bringing to our attention these interesting studies which we cite as [43] and [44], and accordingly we added to paragraph 2:

“Synchronization in the broader sense of coordinating decision-making between humans on complex networks has also been studied [43, 44].”

The paper of Brackbill et. al. focuses on the ability of different groups to achieve the correct answer as a function of the network connectivity. This paper is interesting and is strongly related to our next experiment where we are planning to study the decision making of players in different networks. We added this paper as citation [29] and are referring to it in the introduction:

“...investigating the decision-making processes in different types of groups [25 – 29].”

Reviewer #3 (Remarks to the Author):

“This paper presents an interesting experimental study of synchronization in a human network of interacting violin players. I had reviewed an early version of this paper. The following points remain to be addressed before a publication recommendation can be made.”

1. *“Why is the playing period of each player $T_i(t)$ a function of time? How is it measured exactly?”*

Per comment by reviewer, we added a detailed explanation in the method section how we obtained the $T_i(t)$ as a function of time:

“The output file was analyzed off-line in Matlab by Fourier-transforming the signal in a moving window of 100 ms which allows to identify the different notes and the timing of each note and performed a manual consistency check to verify the results. Next, we calculated the playing period of each player and its location along the musical phrase which is the player phase.”

2. *“It would be better to introduce an equation to define the coupling strength. What does it mean “... and according to a logarithmic scale”?”*

Thanks to the reviewer’s comment, we realized that this statement was misleading. Since we are working at the linear response range of the human hearing we decided to remove the mention of the logarithmic scale to prevent confusion. Therefore, we revised it to:

“The coupling strength is defined as the ratio between the volume of the coupled violin compared to the volume of the player’s own violin while maintaining the total volume that each player hears constant. The volume level was monitored to make sure it stayed inside the linear response range of the human hearing [57].”

3. *“Often the terminology used is not explained as for the cases of “crowd synchrony” or “out-of-phase synchronization”. One cannot expect the broad readership of NC to be familiar with technical terms like these. The other major problem is that complex concepts such as “vortex synchronization” or “arrowhead synchronization” are introduced by looking at figures. I suggest that these states are first defined and then illustrated in figures.”*

Per comment by reviewer, we added citation [19] for explaining the term “crowd synchrony” in paragraph 2:

“Human ensembles and crowd synchrony [19] have been investigated in recent years.”

We added citation [58] for explaining the term “out-of-phase synchronization” and emphasize it in paragraph 10:

“In this out-of-phase synchronization [58], when player i is at the middle of the musical phrase, player j is at the beginning or the end of the phrase, and vice-versa.”

The same citation also elaborates on in-phase state of synchronization so we added it to paragraph 9:

“The delay prevents the players from synchronizing with each other, which leads them to shift from an in-phase synchronization to other states of synchronization [58].”

To elaborate what is the meaning of “vortex synchronization” or “arrowhead synchronization” we revised Fig. 4, to:

And also added explanation to paragraph 11:

“In this stage, we observe two states of synchronization, a vortex state or an arrowhead state. If all the players ignore the same side and follow the other side, they create a vortex state of synchronization where the phase increases monotonically, as seen in Fig. 4(h), while if some players are choosing to follow the player on one side and other players are choosing to follow players on the another side, they create an arrowhead-shaped state of synchronization, as seen in Fig. 4(e).”

4. *“The experiment about increasing the communication delay is interesting. However, I have the following questions: “the players are able to perceive an increase of the communication delay while they are playing. Correct? So they may be able to figure what the actual delay is as time progresses. If yes, then I would argue a control experiment is needed in which the delay is static and does not change in time.”*

We verified that the change in the delay is slow enough so the system remains quasistatic. For this, we measured other values of the delay change-rate and obtained similar results, indicating that the change-rate of the delay is slow enough. We cannot further slowdown the change in the delay to approach the static limit since this would make the entire experiment too long as the players are getting tired when playing more than few minutes. This is explained in paragraph 9: **“We measured coupled violin players when the delay is changing at half the rate, according to $d(t) = 0.0166t$, and obtained similar results. This indicates that, although the delay is constantly changing, the delay change-rate is slow enough so at each point in time the network can be considered as quasistatic. In such a system, the players are not aware to the fact that the delay is changing and only react to its current value.”**

5a. *“One of the main findings is the emergence of an effective network coupling topology for which certain links are neglected/ignored. However, this could be better explained. How is the effective topology determined? Is it time invariant?”*

Per comment by reviewer, we elaborate how we measure the phase of each player at the supplementary materials in the method section:

“The output file was analyzed off-line in Matlab by Fourier-transforming the signal in a moving window of 100 ms which allows to identify the different notes and the timing of

each note, and performed a manual consistency check. Next, we calculated the playing period of each player and its location along the musical phrase which is the player phase. By comparing the phase between two coupled players, we determine if they are following each other, if one is ignoring the other, or if both of them are ignoring each other.”

And we describe the way we determined if player i is following player j in paragraph 9:

“We determined that player i is following player j once their relative phase during at least one musical phrase, follows:”

$$\varphi_j - \varphi_i = 2\pi \frac{d(t)}{T_{ij}(t)}.$$

where $T_{ij}(t)$ is the playing period of players i or j . When Eq. (1) is satisfied, player i is playing in synchrony to player j as it sounds in its earphones. If Eq. (1) is not satisfied, even if the relative phase between them is constant in time, they are not following each other. This can occur when both players are following a third player while ignoring each other.”

The stable topology is not always constant in time which is explained in paragraph 14:

“Once the players found a stable solution they tend to stay in it, while in some cases they switched from one stable solution to another.”

5b. “Are the connections of this topology always bidirectional?”

When a stable state is reached, the connections are bidirectional but before the stable state is reached the connections are not always bidirectional. To elaborate this, we added to the caption of Fig. 3:

“When the delay is zero, the two players are following each other in-phase denoted by a double-head arrow between them. When the delay increases, only one of the player can follow the other which is indicated by a single-head arrow.”

This is also explained in paragraph 14:

“For zero delay in the square lattice configuration, the players are synchronized in phase, and with increased delay, they create vortex states until reaching the state of out-of-phase synchronization, which is a stable solution to the square configuration. In the triangular configuration, the players start with in-phase synchronization, and with increased delay, they cannot find a stable solution, so they ignore some of the connections and change the connectivity of the network to one based on square motifs or open-chains. A network which is based on square motifs or open-chains is following the same dynamics as any chain with even number of players, and thus, can find the highly stable state of out-of-phase synchronization.”

5c. “Also, it appears some of these preferred emerging topology are “square motifs or open chains” -- is there an explanation for this?”

Square motifs as well as open chains have the same dynamics as close chains with even number of players and can thus find the out-of-phase state which is highly stable. We explain this in paragraph 14:

“A network which is based on square motifs or open-chains is following the same dynamics as any chain with even number of players, and thus, can find the highly stable state of out-of-phase synchronization.”

5d. “An alternative counter-explanation for neglecting some of the links, which would find excellent analogies in the studies on coordinated motion of animals, is that the human/animal mind can only keep track of a limited number of “signals” at the same time.”

This is exactly the point of our manuscript, that human mind can only keep track of limited number of signals and therefore, is choosing to ignore some of the links. What we have shown, and the novelty of our study, is the influence of this well-known property of the human mind on the dynamics of complex human networks. To emphasize this, we revised the conclusions to:

“We found that human networks differ from previously studied networks in the ability of each player to adjust its playing period and to change the network connectivity by ignoring a coupled player and effectively deleting the connection. This ability serves as a unique and efficient mechanism to remove frustrating signals that hinder synchronization.”

6. “Therefore, the specific psychology of each player has no role in the macroscopic network dynamics of coupled violin players” not sure psychology is the right word here?”

We agree with the reviewer and revised it to:

“Therefore, the specific strategy each player has for choosing which inputs to follow, has no role in the macroscopic network dynamics of coupled violin players.”

We also revised this sentence in the supplementary materials.

7a. “It’s good that the paper presents a simple model that attempts to explain the experimental observations. However, I do not understand how in the model the three different mechanisms considered for deleting links can all lead to the same dynamics. Do the authors have a clear understanding of this?”

Yes, the different mechanisms are governing the local dynamics, namely which players will be coupled to which, but the global dynamics is identical between all the models. This is due to the fact that a network with square motifs is the only geometry with a stable fixed point of the system and therefore, all models eventually converge to this effective geometry. However, there are large number of possible microscopic solution and in each realization we can obtain a different solution. Per comment by the reviewer, we returned a figure which was absent from this submission showing three representative reduced networks resulting by the three mechanisms. Specifically, we added to the supplementary materials Fig. S4:

Fig. S4. Typical reduced arrays of coupled players situated on triangular arrays resulting when the systems find the out-of-phase synchronization states. Here, we show the connections that the system decided to keep in three representative simulations. We compare three mechanisms for choosing which player to follow and which to ignore: phase - following connections with similar phase; periodicity - following connections with similar periodicity; random - randomly choosing which connections to follow and which to ignore. In all cases, we see that the system finds the out-of-phase synchronization state by reducing the network to network based on motifs with an even number of players, indicating that the specific mechanism for choosing which connection to keep plays no role in the network dynamics.

We choose to present the networks where the square motifs are most obvious these networks are also highly regular. But they are just an example, some simulations lead to networks which were highly ordered and some were chaotic. The square motifs are easier to spot at the highly ordered networks so we presented them.

7b. “Also, how are the frequencies of the Kuramoto oscillators and all the parameters (eg coupling strengths, eta, etc) chosen?”

We chose all the properties to be as similar as possible to the experimental setup. We describe all the properties of the model in the supplementary materials in paragraph 1 section 1:

“In our simulations, the coupling strength was set to $\kappa = 0.2$ and the frequencies were randomly chosen between $\omega = 0.25 \text{ Hz}$ and $\omega = 0.3 \text{ Hz}$ with a uniform distribution, corresponding to a playing period of 3.3 to 4 seconds. Next, we introduce the delay as a function of time according to $d(t) = 0.0332t$ so after 120 seconds the delay reaches 4 seconds.”

We also uploaded all the code to figshare, so it is available to the readers.

7c. *“An alternative model could include a mechanism for network adaptation in which the coupling strengths are tuned in order to maintain synchronization, see Physical Review E 79 (1), 016201 2009 and CHAOS 20 (1), 013103 2010.”*

We thank the referee for bringing to our attention these interesting papers and we read them carefully as well as F. Sorrentino and E. Ott, Phys. Rev. Lett. 100, 114101 (2008) which helped us to understand the model. The papers present a model for analyzing the dynamics of network of coupled chaotic oscillators. The model is based on obtaining an effective potential for every node which governs the dynamic of the network. The main difference between this model and our system is: in our system the players can identify the different inputs, ignore some of them and listen to others, while the model of Sorrentino and Ott is based on the fact that the input of each node is the summation of all its input without the ability to separate them. Perhaps there is a way to extend their model to our system, but it is beyond the scope of this manuscript.

Still, the papers are closely related, and the model presented in them is another evident to the important of the delay on the network dynamics. Therefore, we refer to them in paragraph 3: **“Additionally, the influence of changing the coupling strength or the delay between two individuals is critical for the dynamics of the network [48 – 50] and has not been studied in human networks thus far.”**

8. *“What do the authors exactly mean by “adjust the periodicity”?”*

We thank the reviewer for noting our typo, and we fixed it to:

“We found that human networks differ from previously studied networks **in the ability of each player to adjust its playing period** and to change the network connectivity by ignoring a coupled player and effectively deleting the connection.”

9. *“This sentence from the conclusions is a bit vague: “Our system will be extended to investigate decision making models in different configurations, bifurcation and phase transition in human networks, and the nonlinearity of crowds.”*

Per comment by reviewer, we removed this sentence.